# Identification and Functional Validation of Auxin-Responsive *Tabzip* Genes from Wheat Leaves in Arabidopsis

**DOI:** 10.3390/ijms24010756

**Published:** 2023-01-01

**Authors:** Ziyao Jia, Mengjie Zhang, Can Ma, Zanqiang Wang, Zhonghua Wang, Yan Fang, Jun Wang

**Affiliations:** 1State Key Laboratory of Crop Stress Biology for Arid Areas, College of Agronomy, Northwest A&F University, Xianyang 712100, China; 2State Key Laboratory of Soil Erosion and Dryland Farming on the Loess Plateau, Northwest A&F University, Xianyang 712100, China

**Keywords:** wheat, leaf, auxin response, transcriptome analysis, *TabZIPs*, *TabZIP6D_20*, *Hy5*

## Abstract

Leaves are an essential and unique organ of plants, and many studies have proved that auxin has significant impacts on the architecture of leaves, thus the manipulation of the three-dimensional structure of a leaf could provide potential strategies for crop yields. In this study, 32 basic leucine zipper transcription factors (bZIP TFs) which responded to 50 μM of indole-acetic acid (IAA) were identified in wheat leaves by transcriptome analysis. Phylogenetic analysis indicated that the 32 auxin-responsive *TabZIPs* were classified into eight groups with possible different functions. Phenotypic analysis demonstrated that knocking out the homologous gene of the most down-regulated auxin-responsive *TabZIP6D_20* in Arabidopsis (*AtHY5*) decreased its sensitivity to 1 and 50 μM IAA, while the *TabZIP6D_20/hy5* complementary lines recovered its sensitivity to auxin as a wild type (Wassilewskija), suggesting that the down-regulated *TabZIP6D_20* was a negative factor in the auxin-signaling pathway. These results demonstrated that the auxin-responsive *TabZIP* genes might have various and vital functions in the architecture of a wheat leaf under auxin response.

## 1. Introduction

The leaf is the plant organ for photosynthesizing, and the products of photosynthesis provide more than 90% of the biomass of crops [1]. Wheat (*Triticum aestivum* L.), as one of the most important staple crops in the world, can increase its yield by promoting photosynthetic efficiency by manipulating the architecture of its leaf [1]. The final shape and size of leaves affect the dry matter accumulation of grains; therefore, the whole leaf development stage must be precisely regulated [2]. The growth and development of a leaf is a complicated process involving the interaction of many regulatory factors [3]. Among them, auxin is one of the earliest discovered and most intensively studied phytohormones that affects almost all stages of plant growth and development [4].

The effects of auxin on leaves depend on its concentration. The gradient distribution of auxin concentration is established by local synthesis and polar transportation, which is the crucial factor in regulating plant growth, fruit abscission, and responding to external stimuli including abiotic and biotic stress [5,6,7]. According to previous studies, a moderate increase in auxin concentration stimulates leaf growth, while the super-optimal concentration inhibits this process [8]. During the formation stage of apical-hook development, auxin was asymmetrically accumulated at the concave side of the hook to inhibit cell elongation, and then the differential growth alongside the hook formed the hook finally [5]. In addition, leaf flattening can form wider leaves with better photosynthesis, and the abaxial-enriched distribution of transient auxin conduces to the adaxial-abaxial patterning of the leaf [9]. In general, auxin regulates diverse plant developmental processes through the modulation of auxin-responsive gene expression at the transcriptional level. In the best-characterized auxin-signaling pathway, the cis-regulatory auxin response elements (AuxREs) are bound by correlated auxin response factors (ARFs). Different auxin concentration leads to the different transcriptional activity of ARF, which will significantly affect the transcription of auxin-dependent genes [10,11]. However, recent studies showed that auxin-mediated transcription might also be controlled by other cis-regulatory elements, such as ZREs (for bZIP Response Elements), MREs (for Myb Response Elements) as well as MYC2-related elements [12]. These elements are also dramatically enriched in the region of the promoter of auxin-inducible genes. For example, G-box (CACGTG), a bZIP transcription factor (bZIP TFs) binding site, was highly enriched in ARF6-binding regions [13].

The bZIP family members are recognized by a common region that binds DNA and a leucine zipper dimerization motif, which contributes to their functions as transcriptional factors to control gene activities that involve multiple biological processes of plant growth and development. Previous studies have reported there were 75 *bZIP* genes in *Arabidopsis thaliana* [14], 89 *bZIP* genes in *Oryza sativa* [15], 125 *bZIP* genes in *Zea mays* [16], and more than 176 *bZIP* genes in *Triticum aestivum* [17,18]. Based on their structure and function, all the *bZIP* genes from plants can be divided into 10 subfamilies [18]. The transcription factor long hypocotyl 5 (HY5), a bZIP protein, is the earliest identified and most widely studied transcription factor in promoting photomorphogenesis. Furthermore, studies have demonstrated that nine AUX/IAAs, six ARFs, and six ERFs (for Ethylene-Responsive Factors) were all its target genes, suggesting that HY5 is a signal transducer to link hormone and light signaling [19]. Another study reported that *Arabidopsis* transcription factor bZIP29 participated in leaf and root development by altering the cell number in leaves and in the root meristem [20]. However, the effect of bZIP TFs on the auxin signal pathway during the leaf architecture is still unclear, especially in crop species.

RNA-seq, as one of the powerful tools for transcriptomic analysis, not only provides more information for further studies in the functional genomics of plants but also facilitates an increase in their yield and quality [21,22]. In the present study, we identified 32 auxin-responsive *TabZIP* genes in wheat leaves under the 50 μM IAA by transcriptome analysis and verified their gene expression by qRT-PCR. Then, we analyzed their gene structure, motifs as well as their phylogenetic relationship with other *bZIP*s from *Arabidopsis* and rice, which suggested these auxin-responsive *TabZIPs* could be divided into 8 groups. Finally, we selected the most significantly down-regulated gene, *TabZIP6D-20*, as the candidate gene and verified the gene function of *TabZIP6D-20* in corresponding mutant hy5 and complementary lines *TabZIP6D_20/hy5* in *Arabidopsis*. This study provides basic information on auxin-responsive TabZIP TFs in wheat leaves and their potential application in manipulating the architecture of wheat leaves using transgenic technology.

## 2. Results

### 2.1. Phenotype of Wheat Seedlings under Different Concentrations of IAA

We selected wheat seedlings at the same growth rate (Appendix A) for IAA treatments, and the concentrations of IAA were 0 μM, 1 μM, and 50 μM. After 7 days, the leaves of the wheat seedlings were significantly changed under 50 μM IAA compared with 0 and 1 μM IAA (Figure 1 and Appendix A). Then, we collected the leaves of wheat seedlings that were treated under 0 and 50 μM IAA for transcriptomic analysis.

### 2.2. Identification of Auxin-Responsive TabZIP Genes in Wheat Leaves

A total of 57.34 G nucleotide data, equal to 475,367,674 raw reads and 468,546,670 clean reads were produced from the library (Table 1). The average size of the samples is 78 million reads (11.6 GB), which significantly surpassed the transcript size of the wheat genome that was supposed to contain almost all the transcribed genes within the wheat genome. The minimum Q30 percentage of clean reads was 87.7%, and the content of GC nucleotides was from 54.26 to 57.73%. Furthermore, the percentage of reads mapped to the wheat genome ranged from 92.96 to 95.03%.

A total of 8092 differentially expressed genes (DEGs, Fold Change ≥ 2, and FDR < 0.01) were identified under 50 μM IAA treatment and showed in the volcano plot (every single point represents an auxin-responsive gene). Among them, 3921 were down-regulated DEGs (blue points), and 4981 were up-regulated DEGs (red points) (Figure 2).

There are 6259 DEGs annotated into three aspects of GO (Gene Ontology), including molecular function, cellular component, and biological process. In each aspect of GO, the dominant groups of molecular function were catalytic activity and binding, the dominant groups of cellular component were cell part, organelle, and membrane, and the dominant groups of biological process were the metabolic process and cellular process (Figure 3a). In addition, other groups with specific functions, such as nucleic acid binding transcription factor activity, response to stimulus, and biological regulation were also enriched in the three aspects of GO. Then, we conducted KEGG analysis and found the most dominant pathways were starch and sucrose metabolism, porphyrin and chlorophyll metabolism, the phosphatidylinositol signaling system, mTOR signaling pathway, and circadian rhythm (Figure 3b).

A total of 32 auxin-responsive TabZIP genes were identified from 8092 DEGs, and the heatmap, as well as the hierarchical clustering analysis, was constructed based on their expression level (calculated with log_2_ (FPKM + 1) values, Appendix A) (Figure 4). The results showed that 21 auxin-responsive TabZIP genes were up-regulated, and 11 auxin-responsive TabZIP genes were down-regulated under 50 μM IAA treatment.

### 2.3. Phylogenetic Analysis, Conserved Motif, and Gene Structure Analysis of the Auxin-Responsive TabZIP Genes in Wheat Leaves

In order to understand phylogenetic relationship of the 32 auxin-responsive TabZIP genes with other plant bZIP genes, we performed phylogenetic analysis with 75 AtbZIP genes from *Arabidopsis* (dicotyledonous model plant), 89 OsbZIP genes from rice (monocotyledon model plant), and 196 TabZIP genes from wheat (Appendix A). The results showed that all these bZIP genes were divided into 10 groups, which is consistent with previous reports (Appendix A) [18]. To clearly show the classification of the 32 auxin-responsive TabZIP genes in wheat leaves, we simplified the result of phylogenetic analysis, and they were classified into 8 groups (group Ⅰ to Ⅷ, except for group Ⅸ and Ⅹ) (Figure 5). In addition, we found that the TabZIP genes belonging to group Ⅰ, Ⅱ, Ⅳ, Ⅵ were all up-regulated, while the TabZIP genes from group Ⅶ were all down-regulated, and the rest of the TabZIP genes belonging to group Ⅲ, Ⅴ and Ⅷ were both up-regulated and down-regulated in response to auxin, which implied different roles of the TabZIP genes in the complex auxin-responding network during the development of wheat leaves.

The evolutionary and function closeness of TFs can be revealed by their gene structure and conserved motif, so we applied the Multiple Expectation Maximization for Motif Elicitation (MEME) tool (http://meme-suite.org/, accessed on 7 March 2022) to further analyze the characteristics of the 32 auxin-responsive TabZIPs. Based on the results, 10 conserved motifs were identified, and the 32 auxin-responsive genes were divided into two categories according to their gene sequences that contain motif 1 or motif 2. Except for motif 4, 6, and 7, all were found in either of the two categories; motif 5 and 9 were only found in category 1, and motif 3, 8, and 10 were only found in category 2 (Figure 6a,b), suggesting different combinations of motifs may contribute to the unique function of the 32 auxin-responsive TabZIPs in response to auxin.

The number of introns and the structure of the intron/exon is representative of evolutionary imprints for members in same gene family. As the result showed (Figure 6c), most of the auxin-responsive TabZIP genes have fewer than six introns, and the auxin-responsive TabZIP genes with similar structure, position, and numbers of introns were classified in the same group, which is consistent with the result of phylogenetic analysis. Among the 32 TabZIP genes, TabZIP1B_2 and TabZIP4D_12 have more introns than other TabZIP genes, suggesting they are low conservative due to increased number of evolution events occurring in their evolution process. The intron and motif composition of the 32 auxin-responsive TabZIP genes are registered in Appendix A

### 2.4. Cis-Acting Element Analysis of the 32 Auxin-Responsive TabZIP Genes

To obtain more expression information of the 32 auxin-responsive TabZIP genes, we conducted promoter cis-element analysis and identified numerous cis-elements in their promoter regions (1500 bp upstream of initiation codon) (Appendix A). Based on their annotation, all these cis-elements were classified into four categories, which include hormone-responsive cis-elements (category 1), light responsive cis-elements (category 2), abiotic stresses responsive cis-elements (category 3), and growth and development related cis-elements (category 4) (Figure 7). Meanwhile, we countered the frequency of each cis-elements and found that ABRE (abscisic acid responsive element) and TGA-element (auxin-responsive element) were the top cis-elements in category 1; G-box (light responsive element) was the top cis-element in category 2; LTR (low-temperature responsive), GC-Motif (anaerobic inducible enhancer), ARE (antioxidant response), and DRE (dehydration stress response) were the top cis-elements in category 3; and CAT-box and GT1 motif (related to growth and development) were the top cis-elements in category 4 (Figure 7).

### 2.5. Gene Expression Analysis of Auxin-Responsive TabZIP Genes in Wheat Leaves by qRT-PCR

We performed qRT-PCR to further verify the gene expression level of auxin-responsive TabZIP genes in wheat leaves. Additionally, 20 auxin-responsive TabZIP genes from eight groups were selected and their gene expressions checked under 50 μM IAA treatment, and the results showed that the gene expressions of 16 auxin-responsive TabZIP genes were consistent with RNA-seq data (Figure 8).

### 2.6. Down-Regulated TabZIP6D_20 Is a Negative Regulator of Auxin Response

We cloned the most down-regulated TabZIP6D_20 (Traes_6DS_F8DFD036F) as a candidate gene, which is highly homologous to AtHY5 of *Arabidopsis*, and verified its gene function in hy5 mutant and TabZIP6D_20/hy5 complementary lines (2-3, 3-5 and 4-1) in *Arabidopsis*. The results show that the rosette leaf area of the hy5 mutant was bigger than that of the wild type (Ws) under 0, 1, and 50 μM IAA treatments, indicating the insensitivity of the hy5 mutant in response to auxin. Meanwhile, over-expression of TabZIP6D_20 in the hy5 mutant recovered not only the size of rosette leaves but also the sensitivity to auxin as a wild type (Figure 9a). In addition, leaf width, leaf length, and petiole length of the hy5 mutant all expanded significantly compared with the wild type under 0, 1, and 50 μM IAA treatments, suggesting impaired auxin signaling in the hy5 mutant. At the same time, the TabZIP6D_20/hy5 complementary lines have recovered these phenotypes of the hy5 mutant as wild type (Figure 9b–d). Taken together, the most down-regulated TabZIP6D_20 is a negative regulator of the auxin response.

## 3. Discussion

Auxin is one of the most fundamental plant hormones, which participates in almost all aspects of plant growth and development, and can be used as a powerful tool for manipulating the architecture of plant leaves to promote crop yields. In this study, wheat seedlings were treated with 1 μM and 50 μM IAA, and we found that although the roots of the seedlings were significantly reduced under both concentrations of IAA, the leaves of wheat seedlings were significantly changed only under 50 μM IAA, suggesting different parts of wheat seedlings had different responses to different concentrations of IAA, and 50 μM IAA was a stressful concentration for both parts of wheat seedlings. In addition, the leaves of *Arabidopsis* were also reduced significantly under 1 and 50 μM IAA treatments, indicating a high concentration of auxin inhibited the growth of leaves in both monocots and dicots. In addition, the result of RNA-seq analysis also indicated auxin stress by significantly altering the gene expressions of over 8000 genes as well as activating the biotic and abiotic stress response pathway based on GO and KEGG analysis.

Numerous studies have demonstrated that bZIP family genes have participated in multiple biological processes of plant growth and development, such as seed germination and maturation [23], biotic and abiotic stress response [18], photomorphogenesis [24], floral development [25] as well as phytohormone response [26]. Previous studies have reported that there were 176 TabZIP genes [17] or 191 TabZIP genes [18] in the wheat genome due to different methods and versions of reference genome sequences. In our study, 196 TabZIP genes were identified via the latest version of IWGSC 2.0 and classified into 10 groups. According to the result of phylogenetic analysis and RNA-seq, the 32 auxin-responsive TabZIP genes were found in groups Ⅰ to Ⅷ, except for group 9 and 10. Among them, we found all the members in group Ⅶ were the most significantly down-regulated genes, suggesting their importance in auxin response. In addition, conserved motif and gene structure analysis showed that the 32 auxin-responsive TabZIP genes were divided into two categories with different possible functions; for example, group Ⅲ, Ⅴ, and Ⅵ, which belong to category 1, were related to seed development, vascular development, and pollen development separately; and group Ⅳ and Ⅶ that belong to category 2 were related to photomorphogenesis and anthocyanin accumulation, suggesting auxin-responsive TabZIP genes in group Ⅳ and Ⅶ played an important role in the crosstalk of light signaling and the auxin-responsive pathway. Meanwhile, gene expression of the only one member (TabZIP1B_2) in group Ⅳ was up-regulated (1.16-fold) under 50 μM IAA, and it contained the greatest number of introns (10 introns) in its DNA sequence, suggesting it was a low conservative gene during the evolution. Thus, the members (TabZIP6B_17 and TabZIP6D_20) in group Ⅶ with fewer introns were thought to be more conserved and important genes in the crosstalk of light signaling and the auxin-responsive pathway.

The promoter cis-element analysis showed that light-responsive elements and hormone-responsive elements existed in all 32 auxin-responsive TabZIP genes. Compared with the other 31 auxin-responsive TabZIP genes, the TabZIP6D_20 which belongs to group Ⅶ was found with auxin-responsive element, CAT-box as well as the highest frequency of G-box. Phylogenetic analysis indicated that TabZIP6D_20 has a close relationship with HY5 (AtbZIP56), and previous studies have demonstrated that HY5 participated in photomorphogenesis and anthocyanin accumulation. Therefore, we selected TabZIP6D_20 as a candidate gene for further functional study in *Arabidopsis*. Phenotypic analysis showed that the knocking out of HY5 (the homologous gene to TabZIP6D_20) in *Arabidopsis* mutant hy5 caused an expanded rosette leaf area and longer petiole, while overexpression of TabZIP6D_20 in mutant hy5 recovered these phenotypes as wild type. Meanwhile, compared with the wild type and TabZIP6D_20/hy5 complementary lines, hy5 was more insensitive to exogenous IAA, indicating down-regulated TabZIP6D_20 was a negative regulator in auxin response. It is reported that HY5 was also involved in the convergence between cryptochrome and cytokinin-signaling pathways by interacting with cryptochromes [27]. Other studies demonstrated that the cryptochromes not only participated in blue-light-stimulated photomorphogenesis but also in anthocyanin accumulation [28], cotyledon expansion [29], and hypocotyl elongation arrest [30]. Thus, it could be hypothesized that the HY5 and TabZIP6D_20 may also involve anthocyanin accumulation, cotyledon expansion as well as hypocotyl elongation arrest. Indeed, in our study, we observed the expanded rosette leaf area in mutant hy5, and other studies have demonstrated HY5 could be activated by MPK6-mediated phosphorylation under light and promote the expression of anthocyanin-related genes to enhance anthocyanin accumulation [31]. In addition, Tian et al. reported that there were 22 auxin-responsive TabZIP genes in wheat roots with different roles to different concentrations of IAA; for example, there were 4 TabZIP genes responding to 1 μM IAA, 21 TabZIP genes responding to 50 μM IAA, and 3 TabZIP genes responding to both 1 and 50 μM IAA [17]. In our study, 32 TabZIP genes responding to 50 μM IAA were identified from wheat leaves, combined with 21 TabZIP genes responding to 50 μM IAA that were identified from wheat roots; there were 48 TabZIP genes responding to 50 μM IAA identified altogether in wheat seedlings. In addition, there were five auxin-responsive TabZIP genes that participated in the auxin response in both wheat leaves and roots under 50 μM IAA, suggesting their more important function in the complex network of auxin signaling and different TabZIP genes from different tissues played various roles in response to different concentrations of auxin. Meanwhile, the candidate gene TabZIP6D_20 in our study was found not only to be involved in the auxin response under 1 and 50 μM IAA in wheat roots but also participated in the auxin response under 50 μM IAA in wheat leaves, indicating TabZIP6D_20 was a key TabZIP gene in the auxin-signaling network, and it could be used for manipulating the architecture of the wheat leaf in order to enhance yield and improve seed quality.

In conclusion, in our study, there were 32 auxin-responsive TabZIP genes identified from wheat leaves by RNA-seq, and we verified the function of the most down-regulated auxin-responsive TabZIP6D_20 in mutant hy5, and demonstrated that TabZIP6D_20 was a repressor of the auxin-responsive network. These results provide basic information on auxin-responsive TabZIP genes and their potential application in leaf architecture using transgenic technology, which could enhance yield and improve seed quality in wheat.

## 4. Materials and Methods

### 4.1. Plant Materials, Growth Conditions and IAA Treatments

Materials for *Arabidopsis thaliana* (Brassicaceae): seeds of wild-type Ws (Wassilewskija) and hy5 mutant were obtained from our laboratory. The TabZIP6D_20/hy5 complementary lines were generated as follows: full-length cDNA of TabZIP6D_20 was amplified using high-fidelity Taq enzyme (Code No. DR010, TAKARA) and cloned into the pRI 201-AN-GUS DNA (Code No. 3266, TAKARA) vector. The sequences of primers are listed in Appendix A. The correct recombinant vector was then delivered into Agrobacterium strain GV3101, and the plant transformation was conducted using the floral dip method [32]. At least 3 independent TabZIP6D_20/hy5 complementary lines were obtained from the primary transformants.

Seeds were surface-sterilized with 70% ethanol for 1 min, 1% NaClO for 10 min, rinsed 4 times with distilled water, and planted on 1/2 MS plates (1% sucrose and 0.7% agar). After being cultivated in darkness for 2 days at 4 ℃, the plates were moved to a greenhouse under 24/17 ℃ day/night and 14/10 h light/dark and kept for 5 days, then the seedlings were transplanted to pots with sterilized nutrition soil. After 7 days, the rosette leaves were continuously treated with different concentrations of IAA (0, 1, and 50 μM) for another 7 days and then the rosette leaves were collected, photographed, and measured, including leaf width, leaf length, and petiole length. The information was analyzed in Microsoft Excel 2016 (Microsoft, Redmond, WA, USA), and more than 3 independent biological replicates were conducted in this study.

For wheat (*Triticum aestivum* L.): the seeds of cultivar Chinese Spring were provided by our laboratory. The seeds were sterilized with 70% ethanol for 1 min, 1% NaClO for 15 min, rinsed 4 times with distilled water, placed at 4 ℃ for 2 days, and then moved to a chamber under 24/17 ℃ day/night and 14/10 h light/dark. After 1 day, 30 healthy seedlings with the same growth rate were selected and cultivated in Hoagland’s solution containing different concentrations of IAA (0, 1, and 50 μM). After 7 days, the plants were photographed and leaves from 2 to 3 single plants in each replicate were harvested as 1 biological sample. A total of 3 biological samples with qualified RNA were used for RNA-seq, and more than 3 biological samples were used for qRT-PCR analysis. The lengths of wheat leaves were measured by a ruler, and the data were analyzed in Microsoft Excel (Microsoft, Redmond, WA, USA).

### 4.2. Extracting RNA, Preparing Library and Sequencing

Total RNA was extracted from wheat leaves using the RNA prep Pure Plant Kit (Tiangen, China) according to the manufacturer’s instructions. The concentrations of the extracted RNA were detected using the NanoDrop2000 Spectrophotometer (Thermo Fisher Scientific, Waltham, MA, USA) and the integrity of RNA was tested using Agient 2100, LabChip GX (Perkin Elmer, MA, USA) with 1.2% agarose gel electrophoresis. A total of 1 μg of RNA (RIN value > 7) was applied for library preparation. Sequencing library preparation was conducted according to the manufacturer’s instruction for the VAHTS Universal V6 RNA-seq Library Prep Kit for Illumina.

The isolation and fragmentation of poly (A) + mRNA was conducted using VAHTS^®^ mRNA Capture Beads. The first-strand cDNA was synthesized with 1st Strand Buffer and 1st Strand Enzyme Mix (PCR setting with 25 ℃ for 10 min, 42 ℃ for 15 min, and 70 ℃ for 15 min). Then, second-strand cDNA was synthesized with 2nd Strand Buffer 2 (with dNTP) and 2nd Strand Enzyme Super Mix (PCR setting with 16 ℃ for 30 min and 65 ℃ for 15 min). After repairing both ends and adding a dA tail, the double-strand cDNA was treated with Rapid Ligation Buffer and Rapid DNA Ligase to add adaptors. The cDNA was purified by VAHTS DNA Clean Beads and prepared for library amplification.

Purified and adapted cDNA was treated with PCR Primer Mix and VAHTS HiFi Amplification Mix. Every sample was amplified by PCR for 12 circles (denaturation: 98 ℃ for 10 s, annealing: 60 ℃ for 30 s, and extending: 72 ℃ for 30 s). Then, the production of PCR was purified by VAHTS DNA Clean Beads and validated with Agilent 2100 Bioanalyzer.

The library was tested using Qsep-400 and quantified by a Qubit 3.0 Fluorometer and Qubit TM dsDNA HS Assay (Invitrogen, Carlsbad, CA, USA). Sequencing was performed using PE150 (platform: Illumina NovaSeq 6000 platform, reagent: NovaSeq 6000 S4 Reagent, San Diego, CA, USA).

### 4.3. RNA-seq Reads Assembly and Quantification of Gene Expression

To obtain high-quality clean data, we used Trimmomatic (v0.32) to process the filter data in fastq format to remove technical sequences, which include PCR primers or fragments, adapters, and bases whose quality scores were lower than 20. Then, the obtained high-quality reads were de novo assembled via the String Tie [33]. When redundancy and short contigs from the assembly production were removed, the DEGs were obtained. The gene model annotation files and sequences of a reference genome or proteome about *Arabidopsis* and rice were obtained from online genome databases, such as Ensembl (https://ftp.ensemblgenomes.org/pub/plants/release-42/gff3/triticum_aestivum/, accessed on 14 April 2022), NCBI (https://www.ncbi.nlm. nih.gov/assembly/GCA_900067645.1/#, accessed on 17 April 2022), and UCSC (http://www.ucsc.edu/, accessed on 17 April 2022). Finally, the clean data was aligned with the reference genome using HISAT2 (v2.2.1) software [34]. The number of unique-match reads to RPKM were calculated and normalized to analyze the gene expression. Using the negative binomial distribution model package, DEGs Bio-conductor, differential expression analysis was conducted. The false discovery rate was controlled down using a modified Benjamini and Hochberg method. To detect DEGs, the *p*-value of genes was less than 0.05 [35,36].

### 4.4. GO and KEGG Analysis of DEGs

Based on the annotation results, the official classification of GO, as well as the KEGG database, DEGs were classified by functional and biological pathways and enrichment analysis was conducted using R/clusterProfiler. The calculation method of *p*-value is as follows, and then FDR correction was performed on *p*-value. Generally, a Q value ≤ 0.05 is considered as significant enrichment.
$$P=1-\overset{m-1}{\underset{i=0}{\sum }}\frac{\left(\begin{array}{c} M\\ i \end{array}\right)\left(\begin{array}{c} N-M\\ n-i \end{array}\right)}{\left(\begin{array}{c} N\\ n \end{array}\right)}$$

### 4.5. Phylogenetic Analysis of Auxin-Responsive TabZIP Genes in Wheat Leaf

All protein files of AtbZIPs, OsbZIPs, and TabZIPs were downloaded from the JGI database (https://phytozome.jgi.doe.gov/pz/portal.html, accessed on 27 April 2022) and a multiple sequence alignment was performed using Clustal W program in default parameters. The unrooted neighbor-joining phylogenetic tree was generated using MEGA11 with parameters: 1000 bootstrap tests, Poisson model, and pairwise deletions [37]. The subfamilies of TabZIP genes were according to the topology and bootstrap values of the phylogenetic tree, and 32 auxin-responsive TabZIPs in wheat leaf were marked.

### 4.6. Gene Structure, Conserved Motif and Promoter Cis-Element Analysis of Auxin-Responsive TabZIP Genes in Wheat Leaf

The Gene Structure Display Server 2.0 (GSDS; http://gsds.gao-lab.org/, accessed on 1 May 2022) [38] and the Multiple Expectation Maximization for Motif Elicitation (MEME) tool (http://meme-suite.org/, accessed on 1 May 2022) [39] were used to determine the gene exon-intron organization and identify the protein conserved motifs of the 32 auxin-responsive TabZIP genes, respectively. The optimized parameters used in this study were: maximum number of motifs, 10; the number of repetitions, any; and optimum width of each motif, between 8 and 100 residues. The 1500 bp transcript start site upstream sequence of the 32 TabZIP gene transcript was downloaded from EnsemblPlant (https://plants.ensembl.org/index.html, accessed on 1 May 2022), and the file was submitted to Plant CARE (http://bioinformatics.psb.ugent.be/webtools/plantcare/html, accessed on 2 May 2022) to analyze the promoter enrichment type.

### 4.7. Heatmap on Expression Level of the Auxin-Responsive TabZIP Genes in Wheat Leaf

The MeV program was used to generate the heatmap and hierarchical clustering with the parameters as follows: metric: Pearson uncentred, method: complete. The expression levels of the 32 auxin-responsive TabZIP genes were calculated using transformed data of log_2_ (FPKM + 1) values.

### 4.8. qRT-PCR Analysis of 20 Auxin-Responsive TabZIP Genes in Wheat Leaves

Total RNA was extracted as mentioned above and cDNA was synthesized from 1 μg of total RNA using PrimeScript^®^ (TaKaRa, Shiga-ken, Japan) according to manufacturer’s instruction. The primers used in qRT-PCR were listed in Appendix A, and experiments were performed on the ABI StepOnePlus Real-Time PCR System (Applied Biosystems, Foster City, CA, USA). Each reaction was repeated at least 3 times and the volume of each reaction was 25 μL, including 12.5 μL of SYBR qPCR master mix (TaKaRa, Shiga-ken, Japan), 2.5 μL of diluted cDNA, 1 μL of each primer, and 8μL ddH_2_O. The parameters used in this study were as follows: 95 ℃, 1 min; 40 cycles of 95 ℃, 5 s, 60 ℃, 30 s, and 72 ℃, 45 s. The relative gene expression levels were calculated by the 2^−ΔΔCt method.

## Figures and Tables

**Figure 1 ijms-24-00756-f001:**
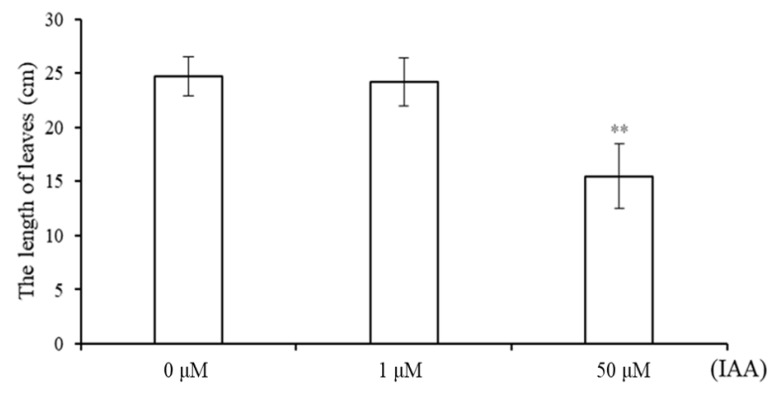
The length of wheat leaves under IAA treatment. Wheat seedlings with the same growth rate were selected and cultivated in Hoagland’s solution containing 0 μM, 1 μM, and 50 μM IAA for 7 days, and then the lengths of wheat leaves were measured and calculated. Double asterisks above bars indicate significant differences between control and treatment groups at *p* < 0.01, *t*-test.

**Figure 2 ijms-24-00756-f002:**
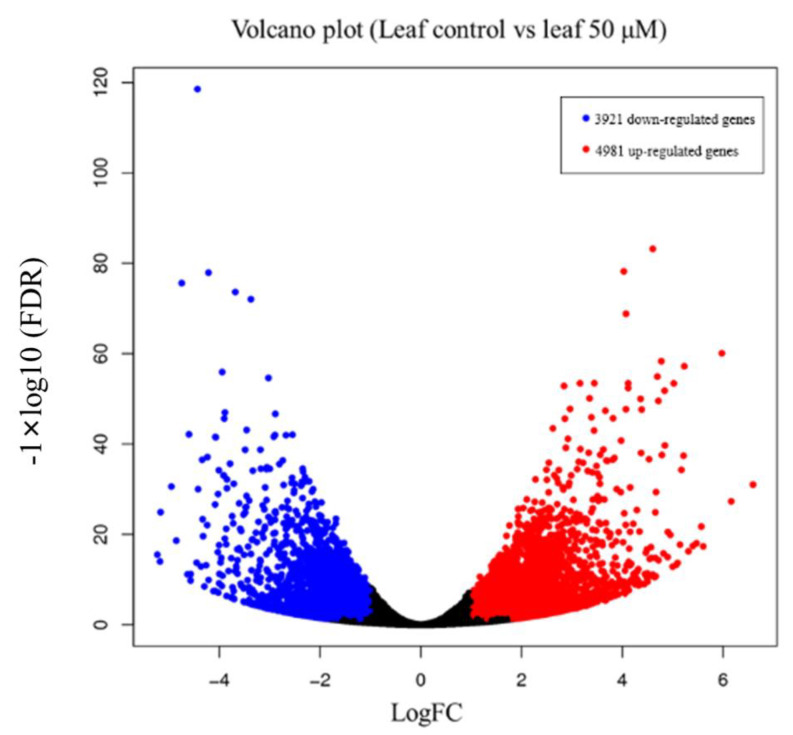
The number of differentially expressed genes (DEGs) shown by volcano plot. Every dot represents a gene. The blue dots represent down-regulated DEGs (3921) with log2 fold-change cut off 2 and *p*-value ≤ 0.05, and the red dots represent up-regulated DEGs (4981) with log2 fold-change cut off 2 and *p*-value ≤ 0.05.

**Figure 3 ijms-24-00756-f003:**
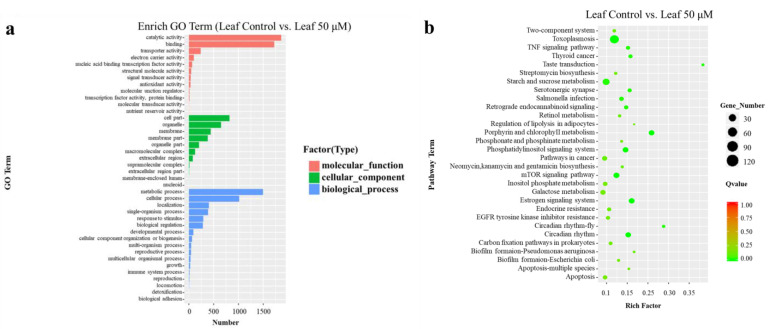
Functional annotation of DEGs based on GO and KEGG database. (**a**) The *X*-axis represents the number of DEGs in the corresponding pathway. The *Y*-axis represents the significantly enriched GO terms (*p* < 0.05). (**b**) The KEGG pathway enrichment scatter map (*p* < 0.05). The *X*-axis (Rich Factor) represents the percentage of DEGs classified to the corresponding pathway in the total gene of the pathway. Related information on KEGG functional classification of DEGs is listed in Appendix A.

**Figure 4 ijms-24-00756-f004:**
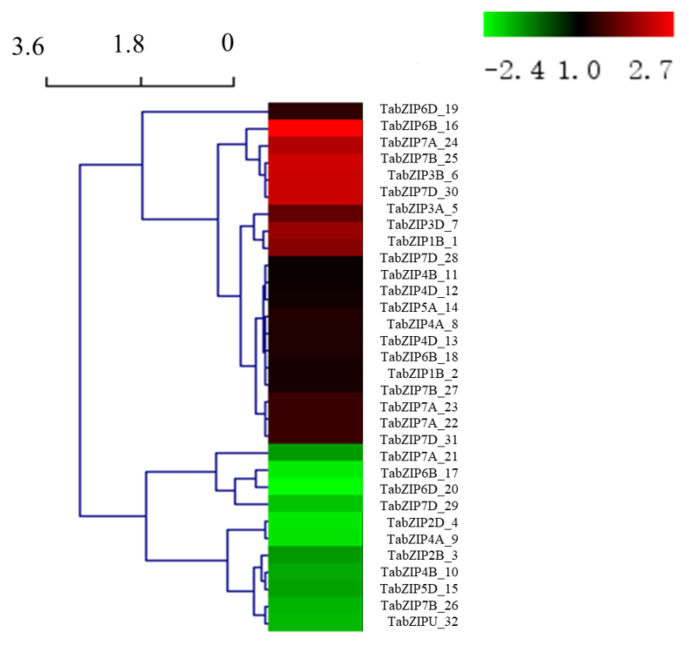
Heatmap for expression levels of 32 auxin-responsive TabZIP genes. The color scale shows increasing expression levels from green to red, which represents log2-transformed FPKM. The expression level is listed in Appendix A.

**Figure 5 ijms-24-00756-f005:**
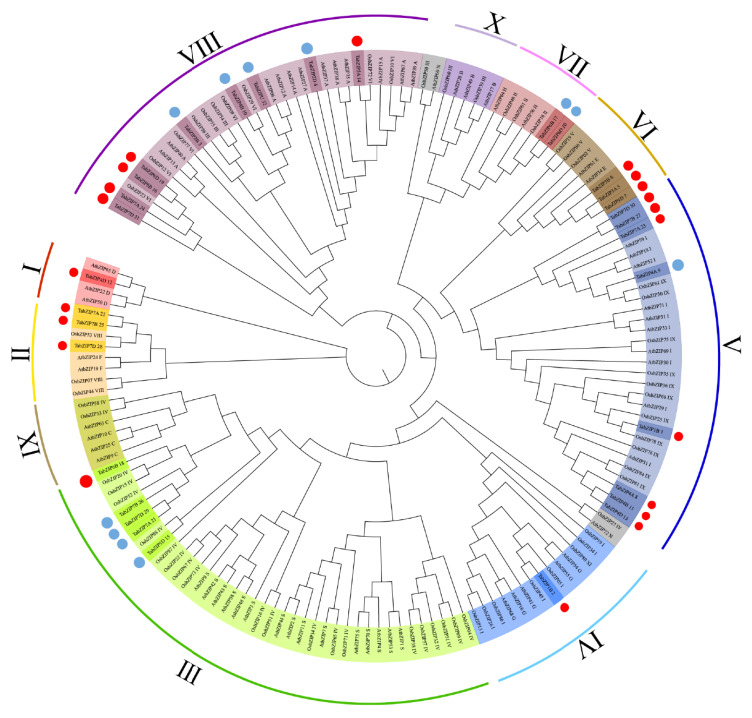
Phylogenetic analysis of 32 auxin-responsive TabZIP genes with other representative bZIP genes from *A. thaliana*, and *O. sativa*. All these bZIP genes were divided into 10 groups (1–10), which were indicated by different colors. The red dots represent 21 up-regulated auxin-responsive TabZIP genes, and the blue dots represent 11 down-regulated auxin-responsive TabZIP genes.

**Figure 6 ijms-24-00756-f006:**
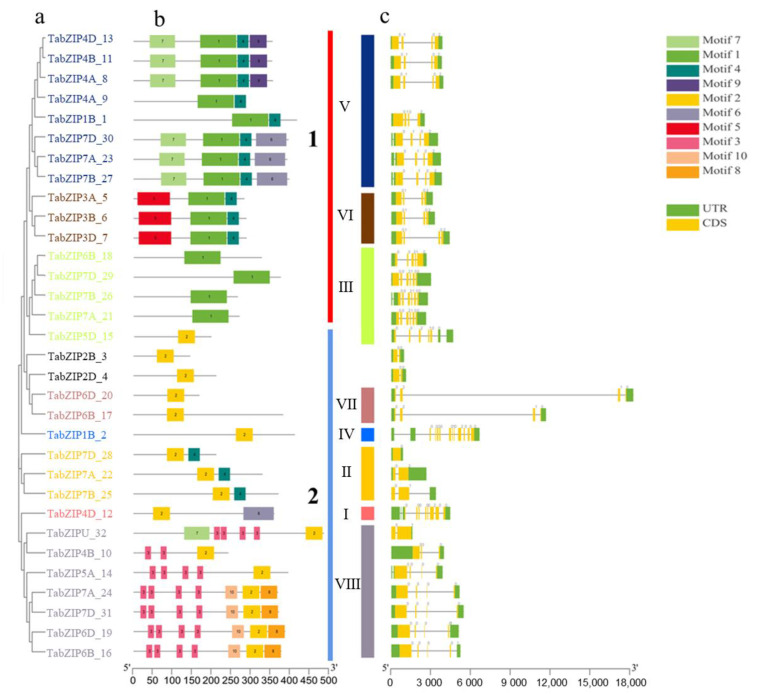
Distribution of introns-exons and conserved motifs of 32 auxin-responsive TabZIP genes in wheat leaves. The phylogenetic relationship (**a**) and location (**b**) of conserved motifs on each gene. (**c**) The distribution of exons/introns of the genes. A total of 32 auxin-responsive TabZIP genes were classified into two categories (the red line represents category 1 and the blue line represents category 2) according to motif 1 and motif 2. A total of 10 conserved motifs were indicated by different colors.

**Figure 7 ijms-24-00756-f007:**
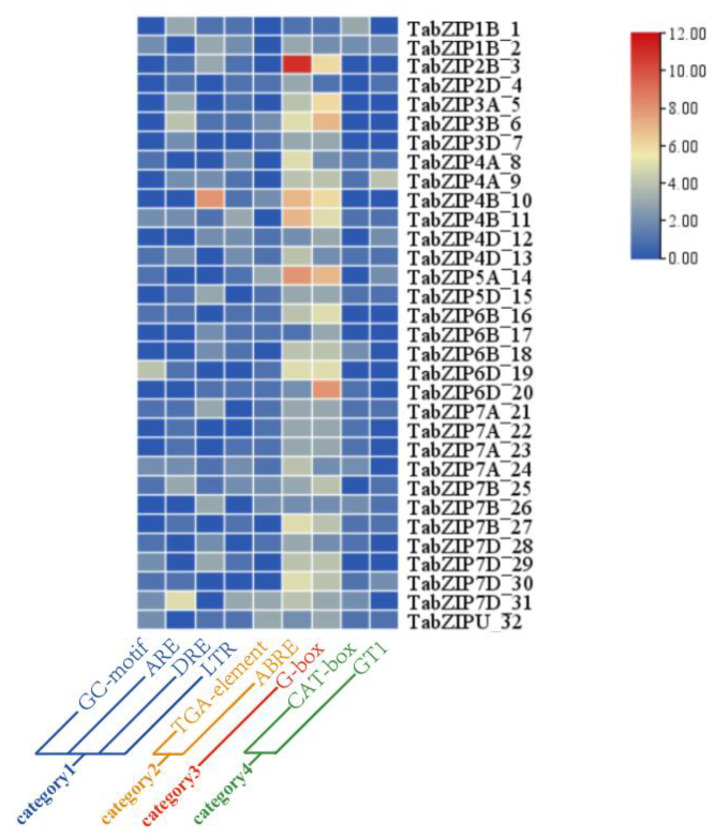
The cis-element of promoters in 32 auxin-responsive TabZIP genes. The color scale represents increasing numbers of the cis-element from blue to red. All these cis-elements were classified into four categories, and nine representative cis-elements from each category were shown.

**Figure 8 ijms-24-00756-f008:**
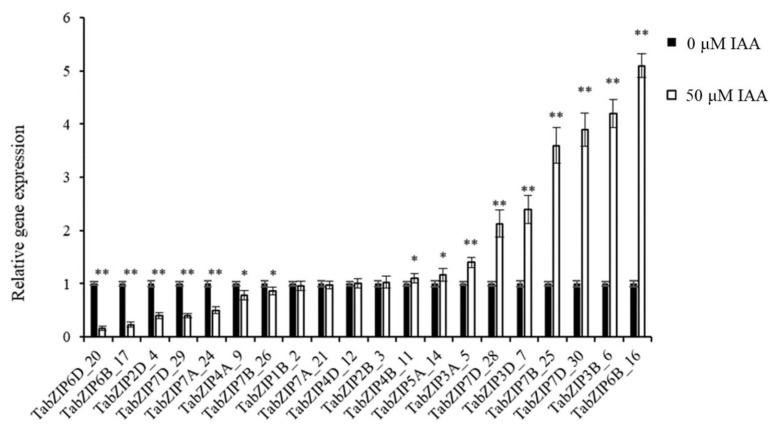
Expression levels of 20 auxin-responsive TabZIP genes by qRT-PCR analysis. Single and double asterisks above bars indicate significant differences between 0 μM and 50 μM treatments at *p* < 0.05 level and *p* < 0.01 level (*t*-test) respectively.

**Figure 9 ijms-24-00756-f009:**
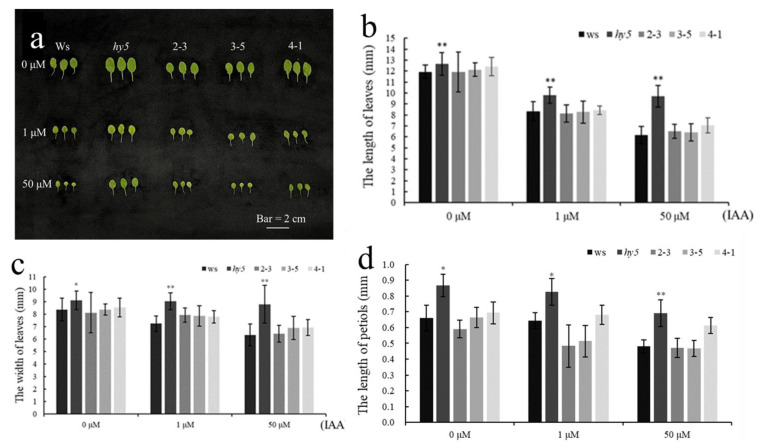
Phenotypic analysis of plant leaves that were treated with different concentrations of IAA for 7-day. (**a**) The phenotype of leaves from the wild type (Ws), hy5, and 35S-TabZIP6D_20/hy5 complementary lines (2-3, 3-5, 4-1). (**b**–**d**) show statistic data of leaf length, leaf width, and petiole length. Fourteen-day-old seedlings were sprayed with different concentrations of IAA (0, 1, 50 μM) for 7 days and then their leaves were photographed and measured. Single asterisks indicated significant differences between wild-type and mutant or complementary lines at *p* < 0.05 level (*t*-test), and double asterisks indicated highly significant differences between wild-type and mutant or complementary lines at *p* < 0.01 level (*t*-test).

**Table 1 ijms-24-00756-t001:** Quality assessment of clean data.

Sample Name	Raw Reads	Clean Reads	Clean Bases	Clean Reads Q20 (%)	Clean Reads Q30 (%)	GC Content of Clean Reads (%)	Total Mapped Rate (%)
0 µM_1	94,625,386	92,792,406	13.74 G	95.72	88.78	55.9	93.74%
0 µM_2	79,896,388	79,079,780	11.76 G	96.72	91.6	57.22	94.52%
0 µM_3	78,477,378	77,536,150	11.53 G	96.47	91.09	57.73	94.20%
50 µM_1	68,767,174	68,035,660	10.12 G	96.63	91.41	55.23	94.04%
50 µM_2	82,892,820	81,892,170	12.16 G	96.75	90.96	55.9	95.03%
50 µM_3	70,708,528	69,210,504	10.23 G	95.24	87.74	54.26	92.96%

The numbers 1–3 after 0 µM and 50 µM identify the three independent biological replicates for the control and auxin treatments, respectively. Q20: The percentage of bases with a Phred value > 20; Q30: The percentage of bases with a Phred value > 30.

## Data Availability

The data presented in this study are available on request from the corresponding author.

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
