# Peer review of "Identification and Functional Validation of Auxin-Responsive Tabzip Genes from Wheat Leaves in Arabidopsis"

_ijms, 2023, doi:10.3390/ijms24010756_

Round 1

Reviewer 1 Report

Jia et al. describe identification of Tabzip genes in wheat that are differentially expressed in leaves under exogenous application of auxin using transcriptome sequencing. Phylogenetic analysis of Tabzip genes is performed together with homologous genes from rice and Arabidopsis. One of the Tabzip genes potentially orthologous to Arabidopsis HY5 gene was functionally characterized by introducing it in Arabidopsis hy5 mutant, where it complemented the non-functional HY5 gene. Overall, the manuscript provides sufficient advance in characterization of Tabzip genes in wheat. Therefore, the manuscript can be accepted for publication in the International Journal of Molecular Sciences, provided the authors address some of the concerns described below. However, reviewer notes that the authors seemingly have divided their data in two different papers – published paper by Tian et al. in IJMS describes auxin response in wheat roots, while the manuscript under review describes auxin response in wheat leaves, but the biological experiment apparently is the same. This needs clarification before publication.

Comments:

  1. While in general manuscript is well written, I believe the discussion section should be improved. First, the observed reduction of leaf length under IAA treatment is not discussed. Is this something expected for monocots? Auxin responses have been studied for ages, so the authors are encouraged to confirm, if the observed phenotypes match what can be expected for monocots under exogenous application of IAA. Second, transcriptome change under auxin treatment has been described in many different plant species including wheat. In this manuscript the authors concentrate only on TabZIP genes, but the rest of the  DEG and pathways are not discussed at all. Are the observed GO categories similar to what can be expected for auxin response in wheat?
  2. None of the supplementary figures are provided, so it is difficult to evaluate leaf phenotypes and how the growth rate of seedlings was evaluated.
  3. Please, use “complement” or “complementary” instead of “compliment”, “complimentary”, e.g., TabZIP6D-20 complements the mutated gene in Arabidopsis hy5 mutant.
  4. Line 55. Please, correct “bind site”.
  5.   Figure 1 and elsewhere. “CK” is not a common abbreviation for “control”. Please, indicate the actual concentration of IAA.
  6. Line 225, 312. Please, rephrase “negative transcription factor”. It could be “negative regulator of auxin response” or “repressor of…”.
  7. There is no Figure 10 (lines 220, 222), so the reference probably is to Figure 9.
  8. Line 239. It is not clear, if authors refer to Tian et al. 2022, when they state that roots were “reduced” under auxin treatment. There is no data on roots in this manuscript. Additionally, they probably mean “root length was reduced”.
  9. Line 271. Correct “were existed”.
  10. It is not clear from Figure 7, how the authors can claim that TabZIP6D-20 gene has the highest frequency of auxin response element, as stated on line 273.

Reviewer 2 Report

I reviewed the paper titled: Identification and Functional Validation of Auxin Responsive Tabzip Genes from Wheat Leaves in Arabidopsis. This study was conducted aiming to provides basic information on auxin responsive TabZIP TFs in wheat leaves and their potential application in manipulating the architecture of wheat leaves by transgenic technology. The objective is interesting for wheat industry, however the mansucript can be accepted if the authors can correct it according to the following comments.

Introduction and discussion is poor and needs to get strength using the relevant literature. I recommend using the following literatures:

Sadat-Hosseini M, Bakhtiarizadeh MR, Boroomand N, Tohidfar M, Vahdati, K. (2020). Combining independent de novo assemblies to optimize leaf transcriptome of Persian walnut. PloS One, 15(4), e0232005.

Khodadadi F, Tohidfar M, Vahdati K, Dandekar AM, Leslie CA. (2020). Functional analysis of walnut polyphenol oxidase gene (JrPPO1) in transgenic tobacco plants and PPO induction in response to walnut bacterial blight. Plant Pathology. 69: 756-764.

Akhlaghi Amiri, N., Asadi Kangarshahi, A., Arzani, K., & Barzegar, M. (2016). Calyx biochemical changes and possibility of reducing thomson orange June drop by nutrition elements and growth regulators. International Journal of Horticultural Science and Technology3(2), 179-186. doi: 10.22059/ijhst.2016.62917

Please clarify in M&M that how many samples have been sequenced? The differential expression analysis without any biological replications is meaningless.

Round 2

Reviewer 1 Report

The authors have adequately revised the manuscript, therefore I recommend it for publication.